# Carcass Composition, Meat Quality and Sensory Quality of Gentile di Puglia Light Lambs: Effects of Dietary Supplementation with Oregano and Linseed

**DOI:** 10.3390/ani11030607

**Published:** 2021-02-25

**Authors:** Giuseppe Scarpa, Simona Tarricone, Marco Ragni

**Affiliations:** 1Ministry of Agricultural, Food and Forestry Policies, Central Inspectorate Department for Quality Control and Agri-Food Fraud Repression, Branch Office of Bari, 70126 Bari, Italy; g.scarpa@politicheagricole.it; 2Department of Agricultural and Environmental Science, University of Bari Aldo Moro, 70125 Bari, Italy; marco.ragni@uniba.it

**Keywords:** lambs, Gentile di Puglia, linseed, oregano, meat quality, fatty acid, sensory quality

## Abstract

**Simple Summary:**

This study characterizes the carcass composition, meat quality, and sensory attributes of an Italian native ovine breed. This investigation aims to implement feeding strategies in light lambs to enhance healthier fatty acids in meat; this is possible by applying extruded linseed and extruded linseed + oregano in a total mixed ration. The dietary treatments have no significant effect on the growth performances, slaughtering data, carcass composition, and quality meat. Moreover, meat chemical composition is not affected by feed supplementation. The diet supplemented with linseed + oregano appears to reduce the concentration of saturated fatty as well as the n-6/n-3 ratio. Due to these changes, meat fatty acid profile is improved, and so are the lamb meat healthy properties. Consequentially, the obtained good sensory traits are compatible with the requirements of the market’s consumer. The combination of nutritional and sensory traits, with properties related to human health that is presented by these native light lambs, qualifies this production as a good choice of red meat to be included in a larger proportion in human food.

**Abstract:**

There is a growing demand by the modern consumer for meat containing less fat and lower levels of saturated fatty acids, which are considered to increase the risk of coronary heart disease. In southern Italy, the Gentile di Puglia breed is one of the most common on farms, and the light lambs are often consumed. The study evaluates the effect of a diet containing extruded linseed (*Linum usitatissimum*) on growth performances, carcass traits, and meat quality in Gentile di Puglia light lambs. Thirty-six male lambs are weaned at about 20 days of age, and divided into three groups—each group is either fed a control diet (C), a diet containing 3% extruded linseed (L), or a diet containing 3% extruded linseed and 0.6% oregano (*Origanum vulgare*) (L + O). The lambs’ growth performances and the slaughtering and dissection data did not differ between groups. Dietary treatments have no significant effect on the quality and chemical composition of *Longissimus lumborum* (Ll). The mount of linoleic acid in Ll meat is significantly higher in the L group, and this positively affected the total content of n-3 Polyunsaturated Fatty Acid, as well as the n-6/n-3 ratio. The good results obtained concerning the sensory traits meet the requirements of the market’s consumer.

## 1. Introduction

The Gentile di Puglia is an Italian ovine breed created in the 15th century from a cross between Merino Spanish rams and “Gentile” ewes reared in some areas of Southern Italy, mainly Apulia and Basilicata [1]. After the Second World War, the consistency of its population exceeded one million heads. At the beginning of the 60′s, the dramatic reduction of its population was determined by many factors [2]. To maintain their market shares, and therefore, their livelihood, Gentile di Puglia breeders began the task of crossing their animals with meat and milk sheep breeds according to their short-term production needs. A large number of animals derived from the Gentile di Puglia converged into a new genetic type named “Merinizzata Italiana”, which contributes to a further decline of the breed. Nowadays, with only 7.200 existing animals, the Gentile di Puglia population is considered endangered according to EU parameters [3,4]. It seems appropriate, in such a situation, to reassess the typical products of the breed and to introduce safeguards to avoid its extinction. In the South of Italy, light lamb’s meat is the main kind of sheep meat appreciated by the consumer. All ruminant’s meat contains a favorable proportion of fatty acids, including the long-chain n-3 fatty acids [5]; nevertheless, lamb and beef meat have been associated with an increased risk of cardiovascular disease due to their high content of saturated fatty acids (SFA) [6]. The formation of large SFA is a result of the biohydrogenation process in the rumen during which bacteria convert unsaturated fatty acids (UFA) to SFA. Therefore, the fatty acids occurring in the rumen are highly saturated and take part in the absorption, as well as deposition of the fat in muscles.

The supplementation of ruminant diets with lipid sources rich in polyunsaturated fatty acids (PUFA) is an effective strategy to improve the nutritional value of meat fat; This diet decrease SFA and enriches PUFA, including the health-enhancing fatty acids (FA), such as conjugated linoleic acid (CLA) and n-3 PUFA [7]. In the last decade, several attempts have been made to increase the PUFA level in meat from livestock species, including dietary supplementation with linseed and linseed oil in lambs [8,9,10,11], kids [12,13], steers [14,15], and pigs [16]. However, increased PUFA level may limit the shelf-life of meat, because they are more prone to oxidation [17]. Antioxidants can be used to prevent or inhibit oxidation; supplementation of ruminants’ diets with natural antioxidants is considered an effective strategy for modulating and improving the fatty acid composition of meat [18], milk, and fresh cheese [19]. Natural antioxidants, such as oregano, sage, thyme, and rosemary, are well accepted by consumers, because they are considered safe and healthy [20,21]. The antioxidant activity in oregano is mainly attributed to carvacrol and thymol [22]; these molecules make the bacterial cell membrane permeable [23], and convert lipids and hydroxyl radicals into stable products [24]. Studies on oregano (*Origanum vulgare*) supplied in lambs’ diet have confirmed its effectiveness in enhancing oxidative stability in meat, as well as in improving its sensory quality [25,26]. Nevertheless, there is little information about the effects of the combination of oregano with linseeds on lamb meat, taking into account that oregano could improve the oxidative stability of meat enriched in unsaturated FA due to the use of linseeds in the diet.

Therefore, the aim of this study was conducted to compare the effect of dietary supplementation with oregano and extruded linseed on growth performances and meat quality of Gentile di Puglia light lambs.

## 2. Materials and Methods

### 2.1. Animal Management and Diet

The study was carried out at the farm “Di Trani” located in Grumento Nova (PZ), Basilicata region, Italy (Latitude 40.281628 N, Longitude 15.896160 E; 650 m asl). It involved 36 male Gentile di Puglia breed lambs. Lambs were reared according to the traditional farming system for the Gentile di Puglia breed as they were exclusively milk-fed, suckling from the ewes until they reached the age of about 20 days. Then lambs were divided into three groups of twelve animals each, homogeneous regarding age (20 ± 3 days old) and initial live weight (9.50 ± 0.25 kg). Each group was assigned to one of the following dietary treatments: C—control group, that received a commercial pelleted feed; L—group fed a pelleted feed containing 3% extruded linseed (*Linum usitatissimum* L.); and LO—group fed a pelleted feed containing 3% extruded linseed and 0.6% dried oregano (*Origanum vulgare* L.) inflorescences. The three pelleted total mixed rations were formulated to be isocaloric and isonitrogenous, and to meet or exceed the nutritional requirements of lambs [27] (Table 1).

The lambs were housed in individual pens (0.8 m^2^/head) with free access to water, and the temperature in pens ranged from 7 to 15 °C. Each animal received feeding ad libitum, and his daily consumption was recorded daily. Each week lambs were weighed to calculate the average daily gain (ADG) until they were slaughtered at 60 days of age; this was done by exsanguination, according to the official veterinary rules after fasting for 12 h, with free water access, and recording the body weight. The hot carcass, skin, fleece, pluck, and full and empty gastrointestinal tract were weighed. Carcasses were hung and chilled at 4 °C (80–82% relative humidity) for 24 h and then weighed. The pH values were measured in the *Longissimus lumborum* (Ll) muscle at the time of slaughter (pH1) and after 24 h (pH24) under refrigerated conditions (4 °C), using a portable instrument (Model HI 9025; Hanna Instruments, Woonsocket, RI, USA) with an electrode (FC 230C; Hanna Instruments) and performing a two-point calibration (pH 7.01 and 4.01).

The refrigerated carcasses were split into two halves by the mid-line; the right side was divided into different cuts (neck, shoulder, leg, steaks, brisket) and weighed separately. The loin was transported from the slaughterhouse to the laboratory under refrigerated conditions. The meat cuts were stored at 4 °C for a further 24 h and then dissected into tissue components (lean, dissectible fat, and bone), and the weight of each tissue was recorded [28].

### 2.2. Chemical Composition of Feed

Representative samples of the pelleted feeds were taken every 20 days, and mixed to obtain a single final pool for each diet, which was then analyzed to determine the chemical composition and fatty acid profile (Table 2). Samples were ground in a hammer mill with a 1-mm screen and analyzed using the following Association of Official Agricultural Chemistry AOAC [29] procedures: Dry matter (method 934.01), ether extract (method 920.39), ash (method 942.05), crude protein (method 954.01), crude fiber (method 945.18), ADF and ADL (method 973.18), and amylase-treated neutral detergent fiber (NDF) (method 2002.04). Metabolizable energy was calculated using INRA system [30].

### 2.3. Physical Parameters of Longissimus lumborum Muscle

On sample muscles meat color and tenderness were determined. Meat color (L* = lightness, a* = redness, b* = yellowness) was determined using a Hunter Lab MiniscanTM XE Spectrophotometer (Model 4500/L, 45/0 LAV, 3.20 cm diameter aperture, 10° standard observer, focusing at 25 mm, illuminant D65/10; Hunter Associates Laboratory Inc., Reston, VA, USA). Three readings were taken for each sample by placing the instrument on different meat areas. The instrument was normalized to a standard white tile before performing the analysis (Y = 92.8, x = 0.3162, and y = 0.3322). The reflectance measurements were performed after the samples were allowed to oxygenate in the air for at least 30 min, to take stable measurements [31]. Three samples (1.25 cm diameter and thick) of each muscle were tested for tenderness by the Warner-Bratzler Shear (WBS) force system using an Instron 5544 testing machine. Shear forces were determined perpendicular to the fiber direction.

### 2.4. Chemical Composition and Fatty Acid Analyses of Longissimus Lumborum Muscle

To analyze the chemical composition of meat, representative sub-samples of Ll muscles were homogenized, and AOAC procedures were used to assess moisture, ether extract, raw protein, ash [29].

Fat was extracted according to the method suggested by Folch et al. [32], using a 2:1 chloroform/methanol (*v*/*v*) solution to determine the fatty acid profile. The fatty acids were then methylated using a KOH/methanol 2N solution [33], and analyzed by gas chromatography (Shimadzu GC-17A) using a silicone-glass capillary column (70% Cyanopropyl Polysilphenylene-siloxane BPX 70 by Thermo Scientific (Pittsburgh, PA, USA), length = 60 m, internal diameter = 0.25 mm, film thickness = 0.25 µm). The starting temperature was 135 °C for 7 min, then it was increased by 4 °C/min up to 210 °C. Samples of each concentrate mixture were used for fatty acid analysis according to the method described above for meat fatty acid profile. Fatty acids were expressed as a percentage (wt/wt) of total methylated fatty acids.

The food risk factors of meat were determined by calculating the Atherogenic (AI) and Thrombogenic (TI) Indices [34]:AI = [(C12:0 + 4 × C14:0 + C16:0)] ÷ [ΣMUFA + Σn-6 + Σn-3];
TI = [(C14:0 + C16:0 + C18:0)] ÷ [(0.5 × ΣMUFA + 0.5 × Σn-6 + 3 × Σn-3 + Σn-3)/Σn-6];
where MUFA are monounsaturated fatty acids.

Lipid oxidation was evaluated in *Longissimus lumborum* muscle samples stored at 4 °C for 48 h after slaughtering by measuring the concentration of 2-thiobarbituric acid reactive substances (T-BARS) [35], and expressed as mg malondialdehyde (MDA)/kg meat.

### 2.5. Sensorial Analysis

The sensorial analysis was performed according to procedures described by Landim et al. [36]. The Ll muscle at the 13th rib was used. Briefly, samples were cut in 1 cm^3^ cube and roasted in a preheated oven to 170 °C. They remained there until the meat temperature was 71 °C at the geometric center of the cube. Forty-one non-trained judges indicated the intensity of sensation on a 9-point scale. Each judge received the meat in plastic containers coded with random three-digit numbers. Between tasting the three samples, each taster received a glass of water at room temperature and a cracker-type biscuit, to remove the residual taste from their mouth. The analysis was based on five sensory descriptors (Table 3); each descriptor was evaluated using a 9-point hedonic scale semi-structured and continuously anchored at extremities with terms that express intensity.

### 2.6. Statistical Analysis

A completely randomized design with three treatments (diets) and twelve replicates (lambs) was used for the study. Data were analyzed using a GLM procedure of SAS software [37] with treatment (diet) as the fixed effect:Y_ij_ = M + A_i_ + E_ij_,
where Y_ij_ = analyzed trait of lambs or meat; M = overall mean; A_i_ = fixed effect of diet; E_ij_ = residual error.

When the diet effect was significant (*p* < 0.05), means were separated and compared by Tukey’s HSD. Significance has been declared at *p* < 0.05; results are reported as least squares mean and their standard error of the mean (SEM).

## 3. Results and Discussion

### 3.1. Performance In Vivo and Post-Mortem of Lambs

No significant differences were found for in vivo performances of lambs (Table 4), diets did not influence lambs’ body weight after 20 and 40 days of trial, but their weight is similar to the weight recorded by Colonna et al. [9]. The ADGs did not show statistical significance; moreover, supplementation of lamb diet with oregano did not show the effect on growth and feed intake; our results are similar to Lestingi et al. [38], even though they used other experimental diets with Gentile di Puglia lambs.

### 3.2. Slaughtering and Carcass Traits of Lambs

Linseed and Linseed + oregano supplementation did not influence the slaughtering data of lambs (Table 5). Other previous studies confirmed that extruded linseed can be an effective ingredient in the diet with no adverse effect on lambs’ or kids’ performance compared to control diet [9,12].

The percentage of the abdominal region on the right half carcass weight was the only section data that was affected by processing treatment (Table 5): C lambs recorded the highest abdominal region percentage compared to L + O lambs (*p* = 0.034).

No effect of feeding extruded linseed and oregano was observed on the main meat cuts; however, the lean proportions of L + O lambs’ leg (Table 6) reported the highest percentage of lean (*p* = 0.026), while showed a lower percentage of dissectible fat (*p* = 0.047) in comparison to the L group. Previous studies evaluating different natural dietary supplements have shown variable results concerning carcass traits [39,40].

### 3.3. Physical Characteristics of Longissimus Lumborum Muscle of Lambs

Physical characteristics determined on Ll muscles are shown in Table 7; muscle pH at 24 h post-mortem was affected by diet supplementation showing higher values (*p* = 0.023) on the L + O group compared to the C group (5.47 vs. 5.32). All pH_24_ value recorded in this trial was lower than pH founded by Marcon et al. [41], Pena-Bermudez et al. [42], and Rant et al. [43]. Meat color features were quite similar among the three supplementations studied, except for the a* index where the L group showed (*p* = 0.049) the lowest red value compared to the L + O group (10.42 vs. 9.41), while Rant et al. [43] and Marino et al. [44] recorded the intermediate value.

T-BARS values (mg of malondialdehyde—MDA/kg meat) showed were similar for Linseed and Linseed + oregano group; this result can explain the use of linseed, which makes the meat easier to oxidize [42].

### 3.4. Chemical Composition of Longissimus Lumborum Muscle of Lambs

Dietary treatments had no significant effect (*p* > 0.05) on *Longissimus lumborum* chemical composition (Table 8), as Fusaro et al. [45] didn’t show the effects of dam dietary treatment on the chemical composition of the suckling lamb *Longissimus lumborum*.

### 3.5. Fatty Acid Composition of Longissimus Lumborum Muscle of Lambs

Table 9 shows the effect of lamb dietary treatment on the FA composition of lamb *Longissimus lumborum*. The results showed that linseed supplementation lowers SFA levels (*p* = 0.039), especially C14:0 (*p* = 0.005), while L + O supplementation showed a lower value of C12:0 (*p* = 0.044) compared to C, which is desirable because these FA are hypercholesterolemic and are associated with a higher risk of cardiovascular disease and type 2 diabetes [46,47]. Our results are consistent with those of Fusaro et al. [45] and Miltko et al. [46], who observed a greater decrease in SFA concentrations in intramuscular fat of lambs sucking ewe that received diet supplementation with linseed oil than in the control diet. On the other hand, Giannico et al. [8], shows an increase of SFA level in the Gentile di Puglia intramuscular fat of lambs fed with linseed oil and extruded linseed integration compared with lambs fed with soybean oil.

Dietary treatments had a not significant effect on *Longissimus lumborum* MUFA composition; only C16:1 showed a lower value (*p* = 0.048) in intramuscular fat of L lambs in comparison to C lambs (0.56% vs. 1.58%). Our results are in agreement with those reported by other authors [8,44,45,48,49].

In our experiment, the lambs in groups L and L + O exhibited higher percentages of linoleic acid (*p* = 0.005), α-linolenic acid (*p* = 0.001), and eicosapentaenoic acid—EPA (*p* = 0.003) than the animals in group C and the same results are showed by Giannico et al. [8] and Fusaro et al. [45] papers. The conversion capacity of α-linolenic to health-promoting long-chain n-3 PUFA is limited in humans [47], which reinforces the significance of its dietary supply. The increase in the concentration of long-chain n-3 PUFA coincided with an increase in dietary C18:3 n-3 concentration of linseed rations.

The meat FA profile was similar to the FA composition of feed, and it was not surprising that changes in feed FA composition can induce significant differences in the FA profile of meat and fat deposit in young lambs. In the young lambs, the essential FAs were incorporated directly into the muscle rather than being stored in the adipose tissue, which is considered an important metabolic role [45].

The n-6/n-3 ratio is used to evaluate the nutritional value of fat for human consumption, and this ratio is strongly dependent on the FA profile of the ration fed to ruminants. This ratio is particularly beneficial in the meat of ruminants that consume forages or oilseed with an increased C18:3 content [47]. Lowering the n-6/n:3 coefficients in food production has been recommended to prevent or modulate certain human diseases, and it should range between 1 and 4 [50]. Our results were within this range; the n:6/n:3 ratios found in the L and L + O groups are the lowest. The same results are showed by Gómez-Cortés et al. [51] and Rotondi et al. [12] that found a lower n-6/n-3 ratio in the meat of animal fed linseed.

The indices of atherogenicity and of thrombogenicity are indicators assessing the level and the interrelation of some fatty acids that have effects on the occurrence of coronary heart diseases [33]. In this study, the meat of control group lambs showed a markedly greater atherogenic (*p* = 0.041) and thrombogenic (*p* = 0.039) index compared to the other two groups. The same effect of linseed in the diet is showed by Fusaro et al. [45], and also their results are not statistically significant.

### 3.6. Sensory Analysis of Longissimus Lumborum Muscle of Lambs

Sensory analysis results of Ll muscle in Gentile di Puglia lambs, according to feed, are shown in Table 10. Several studies show that the taste of sheep meat is influenced by the animal’s diet; meat from animals fed with concentrate shows higher flavor intensity than that of grazing sheep [52,53,54]. We found similar results in this study; the linseed and oregano supplementation influenced the judges’ evaluation. The meat from the L + O lambs were tender (*p* = 0.007) and juicy (*p* = 0.006) then the meat of lambs of the other two group; while L + O meat was succulent (*p* = 0.008) then the C lambs’ meat, it was also more flavorful (*p* = 0.048) compared with L meat (5.56 vs. 4.56).

For overall acceptability, the sum of descriptors contributing to the acceptance of the lamb meat ranged from 7.20 to 7.32, indicating that the meat has good acceptance.

## 4. Conclusions

This paper contributes to a better understanding of the meat quality and potential health benefits of FA profiles in native Italian sheep breeds fed with linseed supplementation. According to the results of the present study, linseed and linseed + oregano supplementation did not influence growth and feed intake of lambs, or their slaughtering and carcass traits. These supplementations did not change physical characteristics or chemical composition of lambs’ meat.

As expected, FA profile in Ll muscle was affected by linseed and linseed + oregano supplemented to lamb diets, in fact, L and L + O meats showed the lowest value of healthy parameters, such as the n-6/n-3 ratio, atherogenic and thrombogenic indexes. Additionally, the supplementation of linseed to diets seems to reduce the concentration of lauric and myristic acid and the total SFA; instead, linseed supplementation increase the concentration of total n-3 PUFA, particularly α-linolenic and EPA. Linseed and oregano tended to reduce the SFA percentage, significantly increased the PUFA and UFA concentration, particularly the linoleic acid, α-linolenic, and total n-6 PUFA.

Linseed and oregano supplementation to the diet of lambs improve meat flavor and overall acceptance by consumers, as has been seen in succulence, tenderness, and juiciness valuation.

These positive results in terms of potential benefits for human health and consumers’ acceptance of Gentile di Puglia lambs may represent an opportunity for valorization and promotion of this breed. Further trials will be conducted with the use of other officinal plants rich in antioxidants.

## Figures and Tables

**Table 1 animals-11-00607-t001:** Composition of the experimental diets.

Ingredients (% As-fed Basis)	Dietary Treatment ^1^		
	C	L	L + O
Corn	31.00	31.00	30.40
Faba bean	10.00	8.50	8.50
Wheat bran	10.00	10.00	10.00
Barley	9.00	9.00	9.00
Wheat flour shorts	9.00	9.00	9.00
Sunflower meal	8.00	7.50	7.50
Dehulled soybean	6.00	6.00	6.00
Sugar beet pulp	6.00	6.00	6.00
Soybean hulls	4.00	4.00	4.00
Molasses	3.00	3.00	3.00
Vitamin-mineral premix	3.00	3.00	3.00
Soybean oil	1.00	-	-
Extruded linseed	-	3.00	3.00
Oregano (dried inflorescences)	-	-	0.60

^1^ C, control feed; L, control feed +3% extruded linseed; LO, control feed +3% extruded linseed +0.6% oregano.

**Table 2 animals-11-00607-t002:** Chemical and fatty acid composition of diets.

Variable	Dietary Treatment ^1^		
	C	L	L + O
Chemical composition (% on DM basis)			
Crude protein	15.51	15.60	15.61
Ether extract	3.66	3.70	3.71
Ash	3.41	3.49	3.56
Crude fiber	7.91	7.92	8.32
NDF ^2^	21.19	21.24	21.15
ADF ^2^	9.58	9.56	9.53
ADL ^2^	1.79	1.86	1.85
ME (MJ)	10.16	10.18	10.05
Fatty acid composition (% FA methyl esters)			
C16:0 (palmitic)	9.23	7.47	7.39
C18:0 (stearic)	1.18	3.55	4.08
C18:1 n-9, *cis* 9 (oleic)	17.78	18.76	17.99
C18:2 n-6 (linoleic)	15.16	22.15	20.42
C18:3 n-3 (α-linolenic)	4.65	31.00	30.68
C22:5 n-3 (DPA)	0.46	0.17	0.27
C22:6 n-3 (DHA)	0.29	0.28	0.28

^1^ C, control feed; L, control feed + 3% extruded linseed; LO, control feed + 3% extruded linseed + 0.6% oregano. ^2^ NDF, neutral detergent fiber; ADF, acid detergent fiber; ADL, acid detergent lignin.

**Table 3 animals-11-00607-t003:** Descriptors used in the quantitative descriptive sensory analysis of lamb meat.

Descriptor	Definition
Lamb flavor ^a^	Mixed experience of olfactory, gustatory, and tactile sensations perceived during the tasting. Flavor intensity of cooked lamb.
Succulence ^b^	The first perception of the quantity of liquid liberated by the sample of meat in the mouth.
Tenderness ^c^	The force required to compress a piece of meat between the molar teeth, evaluated at the first bite
Juiciness ^d^	Perception of the amount of liquid released from the meat sample in the mouth after fifth bite.
Overall acceptance ^e^	Sum of quality attributes that will contribute to determining the degree of product acceptance by panelists.

^a^ 0 = not detected, 9 = very intense; ^b^ 0 = extremely dry, 9 = extremely succulent; ^c^ 0 = very tough, 9 = very tender; ^d^ 0 = very dry, 9 = very juicy; ^e^ 0 = very bad, 9 = very good.

**Table 4 animals-11-00607-t004:** Effect of diet on in vivo performances of lambs *.

Item	Dietary Treatment ^1^			SEM ^2^	*p*-Value
	C	L	L + O		
Initial—20 d BW ^3^ (kg)	9.39	9.81	9.42	0.923	0.911
40 d BW (kg)	13.86	13.93	13.91	1.557	0.335
60 d BW (kg)	19.53	19.03	19.30	2.220	0.125
Average daily BW gain 20–40 (kg/d)	0.223	0.206	0.224	0.044	0.074
Average daily BW gain 40–60 (kg/d)	0.258	0.232	0.245	0.054	0.985
Average daily BW gain 20–60 (kg/d)	0.241	0.219	0.235	0.042	0.051
Feed conversion ratio	4.32	4.17	4.64	0.202	0.061

* results are reported as least squares mean; ^1^ C, control feed; L, control feed +3% extruded linseed; LO, control feed +3% extruded linseed +0.6% oregano. ^2^ SEM: Standard error of means. ^3^ BW: Body Weight.

**Table 5 animals-11-00607-t005:** Effect of diet on slaughtering data of lambs *.

Item	Dietary Treatment ^1^			SEM ^2^	*p*-Value
	C	L	L + O		
Final BW (kg)	19.65	19.13	19.30	2.234	0.564
Slaughter weight (kg)	18.82	18.51	18.75	2.099	0.432
PV netto (kg)	17.44	16.85	17.34	1.992	0.617
Skin + fleece (%)	13.13	13.24	13.84	0.867	0.265
Hot Carcass dressing (%) ^3^	67.72	67.09	67.94	2.457	0.745
Cold Carcass dressing (%) ^3^	65.73	64.75	65.82	2.800	0.889
Right half carcass (kg)	11.60	11.15	11.75	1.524	0.513
Meat cuts (%) ^4^					
Neck	5.77	6.01	6.04	0.463	0.078
Shoulder	15.34	15.83	15.41	0.785	0.124
Leg	25.67	26.33	26.38	1.040	0.525
Steaks	12.59	12.60	12.12	0.666	0.105
Abdominal region	4.19 ^a^	3.86 ^ab^	3.77 ^b^	0.343	0.034
Loin	5.72	5.89	5.85	0.373	0.066
Brisket	7.83	7.65	7.69	0.407	0.071
Offal	8.69	8.23	8.16	0.930	0.494

* results are reported as least squares mean; ^1^ C, control feed; L, control feed +3% extruded linseed; LO, control feed +3% extruded linseed +0.6% oregano. ^2^ SEM: Standard error of means. ^3^ % on body weight; ^4^ % on half carcass weight, means with different letters within each row differ significantly: a, b: *p* < 0.05.

**Table 6 animals-11-00607-t006:** Dissection data (% on cut weight) of leg and loin of lambs *.

Item	Dietary Treatment ^1^			SEM ^2^	*p*-Value
	C	L	L + O		
Leg (kg)	0.66	0.69	0.65	0.133	0.085
Lean (%)	48.24 ^b^	48.04 ^b^	54.27 ^a^	3.953	0.026
Dissectible fat (%)	15.96 ^ab^	16.42 ^a^	12.88 ^b^	4.219	0.047
Bone (%)	35.80	35.55	32.84	5.005	0.085
Loin (kg)	2.96	2.90	3.07	0.415	0.546
Lean (%)	66.88	68.34	68.69	1.866	0.081
Dissectible fat (%)	9.43	8.00	7.71	1.805	0.149
Bone (%)	23.69	23.66	23.59	1.466	0.142

* results are reported as least squares mean; ^1^ C, control feed; L, control feed + 3% extruded linseed; LO, control feed + 3% extruded linseed + 0.6% oregano. ^2^ SEM: Standard error of means. Means with different letters within each row differ significantly: *p* < 0.05.

**Table 7 animals-11-00607-t007:** Meat characteristics from *Longissimus lumborum* muscle *.

Item	Dietary Treatment ^1^			SEM ^2^	*p*-Value
	C	L	L + O		
pH_1_ ^#^	6.56	6.53	6.55	0.159	0.052
pH_24_ ^##^	5.32 ^b^	5.40 ^ab^	5.47 ^a^	0.089	0.023
L*	41.40	41.86	40.26	2.771	0.761
a*	9.73 ^ab^	9.41 ^b^	10.42 ^a^	1.013	0.049
b*	11.67	11.36	11.37	1.279	0.963
WBS, kg/cm^2^	2.03	2.40	2.19	0.534	0.776
T-BARS (mg MDA/kg meat)	0.318	0.382	0.371	0.122	0.066

* results are reported as least squares mean; ^1^ C, control feed; L, control feed +3% extruded linseed; LO, control feed +3% extruded linseed +0.6% oregano. ^2^ SEM: Standard error of means. ^#^ pH_1_ at 1 h post-mortem; ^##^ pH_24_ at 24 h post-mortem; L*, Lightness; a*, redness; b* yellowness; WBS, Warner–Bratzler shear force; means with different letters within each row differ significantly: a, b: *p*< 0.05.

**Table 8 animals-11-00607-t008:** Chemical composition (%) from *Longissimus lumborum* muscle *.

Item	Dietary Treatment ^1^			SEM ^2^	*p*-Value
	C	L	L + O		
Moisture	75.06	75.67	74.91	0.638	0.084
Protein	19.17	19.05	19.55	0.652	0.070
Lipid	3.68	3.60	3.50	0.838	0.055
Ash	1.13	1.12	1.15	0.067	0.095

* results are reported as least squares mean; ^1^ C, control feed; L, control feed +3% extruded linseed; LO, control feed +3% extruded linseed + 0.6% oregano. ^2^ SEM: Standard error of means.

**Table 9 animals-11-00607-t009:** Fatty acid composition (% total FA methyl esters) of lamb meat from *Longissimus lumborum* muscle *.

Item	Dietary Treatment ^1^			SEM ^2^	*p*-Value
	C	L	L + O		
Total Fatty acids (g/100 g muscle)	3.26	3.18	3.10	0.241	0.304
C10:0 (capric)	0.37	0.36	0.31	0.220	0.256
C12:0 (lauric)	3.52 ^a^	3.00 ^ab^	2.87 ^b^	0.592	0.044
C14:0 (myristic)	3.42 ^A^	2.33 ^B^	2.74 ^AB^	0.758	0.005
C15:0	0.07	0.08	0.05	0.115	0.074
C16:0 (palmitic)	23.92	23.16	22.00	2.850	0.075
C17:0	0.56 ^B^	1.56 ^Aa^	0.80 ^b^	0.415	0.003
C18:0 (stearic)	14.94	13.96	14.81	1.492	0.063
C20:0	1.07	1.81	1.46	0.613	0.535
∑ SFA	50.99 ^a^	48.70 ^b^	47.38 ^b^	2.279	0.039
C14:1	0.50	0.69	0.45	0.492	0.456
C15:1	0.08	0.09	0.10	0.076	0.189
C16:1 n7 (palmitoleic)	1.58 ^a^	0.56 ^b^	1.52 ^ab^	0.326	0.048
C17:1	0.17	0.28	0.14	0.160	0.401
C18:1 n9 *trans* (elaidic)	3.16	3.02	3.31	1.444	0.155
C18:1 n9 *cis* (oleic)	34.71	34.72	34.60	2.461	0.087
∑ MUFA	40.48	39.43	40.62	2.801	0.062
C18:2 n6 c9 c12 (linoleic)	5.99 ^B^	7.82 ^A^	8.06 ^A^	1.124	0.005
CLA c9, t11	0.86	0.97	0.87	0.229	0.218
CLA t10, c12	0.32	0.34	0.39	0.238	0.165
C18:3n3 (α-linolenic)	0.41 ^B^	0.70 ^A^	0.71 ^A^	0.071	0.001
C20:4 n6 (ARA)	0.17	0.12	0.13	0.073	0.316
C20:5 n3 (EPA)	0.09 ^B^	0.21 ^A^	0.19 ^A^	0.024	0.003
C22:5 n3 (DPA)	0.01	0.05	0.13	0.186	0.240
C22:6 n3 (DHA)	0.30	0.50	0.62	0.410	0.155
∑ PUFA	8.53 ^B^	11.86 ^A^	11.99 ^A^	1.431	0.007
∑ UFA	49.01 ^b^	51.30 ^ab^	52.62 ^a^	2.279	0.032
n-6	7.32 ^B^	9.14 ^A^	9.45 ^A^	1.075	0.001
n-3	0.90 ^b^	2.26 ^a^	1.85 ^ab^	1.059	0.034
n-6/n-3	8.70 ^A^	5.24 ^B^	5.51 ^B^	2.152	0.002
A.I.	0.85 ^a^	0.70 ^b^	0.69 ^b^	0.113	0.031
T.I.	1.58 ^a^	1.29 ^b^	1.29 ^b^	0.218	0.029

* results are reported as least squares mean; ^1^ C, control feed; L, control feed +3% extruded linseed; LO, control feed +3% extruded linseed +0.6% oregano. ^2^ SEM: Standard error of means. SFA—saturated fatty acids (sum of C10:0 + C12:0 + C14:0 + C15:0 + C16:0 + C17:0 + C18:0 + C21:0 + C22:0 + C24:0); MUFA—monounsaturated fatty acids (sum of C14:1 + C15:1 + C16:1 c9 + C17:1 c10+ C18:1 t11 + C18:1 t9 + C18:1 t10 + C18:1 c9 + C20:1 + C24:1); Total n-6 (sum of C18:2 c9;c12 + C18:2 c9; t11 + C18:3 + C20:3 +C20:4); Total n-3 (sum of C18:3 + C20:3 + C20:4 + C20:5 + C22:6); PUFA—polyunsaturated fatty acids (sum of n-6 +n-3); means with different letters within each row differ significantly: a, b: (*p* < 0.05); A, B: (*p* < 0.01).

**Table 10 animals-11-00607-t010:** Sensory analysis of meat from the *Longissimus lumborum* muscle *.

Item	Dietary Treatment ^1^			SEM ^2^	*p*-Value
	C	L	L + O		
Judge (*n*)	41	41	41		
Meat Flavor	4.90 ^ab^	4.56 ^b^	5.56 ^a^	2.243	0.048
Succulence	4.80 ^B^	5.21 ^AB^	5.87 ^A^	1.788	0.008
Tenderness	6.95 ^B^	6.82 ^B^	7.73 ^A^	1.548	0.007
Juiciness	6.22 ^B^	6.37 ^B^	7.41 ^A^	1.506	0.006
Overall acceptance	7.21	7.20	7.32	1.196	0.078

* results are reported as least squares mean; ^1^ C, control feed; L, control feed +3% extruded linseed; LO, control feed +3% extruded linseed +0.6% oregano. ^2^ SEM: Standard error of means; means with different letters within each row differ significantly: a, b: (*p* < 0.05), A, B: (*p* < 0.01).

## Data Availability

All data generated or analyzed during this study are included in this article.

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
