# Peer review of "Carcass Composition, Meat Quality and Sensory Quality of Gentile di Puglia Light Lambs: Effects of Dietary Supplementation with Oregano and Linseed"

_animals, 2021, doi:10.3390/ani11030607_

Round 1

Reviewer 1 Report

General comments

The research object of this manuscript is not certainly original, but represents a valuable contribution to confirm as linseeds are an effective source for improving the health value of meat fat.

However, there are several concerns that have to be solved. The main criticism regards English grammar and language that need to be carefully revised.

Specific remarks and suggestions

L12. “sensory quality” should be substituted by “sensory attributes”.

L17. “was not affected” instead of “has not affected”.

L21-23. This sentence should be rearranged: “The combination of nutritional and sensory traits with properties related to human health….”.

L61. …because they are… (“of” has to be removed).

L67. This assertion means that the authors hypothesize that the effect of oregano supplementation could be different in the meat from Gentile di Puglia lambs than in meat from lambs of other breeds. A more convincing motivation could be that there is few information about the effects of combination of oregano with linseeds on lamb meat, taking into account that oregano could be able to improve oxidative stability of meat enriched in unsaturated FA due to the use of linseeds in the diet.

L72. …the dramatic reduction of its population…

L88. The study was carried out….

L91. The colon after “as” has to be removed.

L92. Were the lambs assigned to the treatments in a homogeneous way? This aspect has to be reported.

L126. Net energy is more suitable for ruminant species.

Table 2. The sum of fatty acids of C diet is lower than 50%. What about the other fatty acids? Were DPA and DHA present in the diets at these levels?

L171-176. The use of a 9-point hedonic scale is repeated more times.

L190.  …diets did not influence… (the use of “no” with a verb, here and in many cases in the text, is not correct).

L191-193. The comparison with Leal et al. seems not to be particularly pertinent, it does not explain the lack of significant differences in the lambs ADG. Whereas in the comparison with Lestingi et al., the authors should specify that lambs of the same Gentile di Puglia breed were used also in that research.

Table 4. What about the feed intake of lambs? Although it can be calculated by the feed conversion ratio (kg/d has to be removed), the statistical comparisons could be showed. In addition, the “p value” is never higher than 1; the levels of significance >1 have to be checked also in the tables 5 and 7.

Table 5. The letters used for means comparisons have to be reported for all three compared means (see abdominal region). Also the other tables have to be checked for this aspect.

L214. This sentence should be moved in the previous paragraph.

Table 6. A note for pH1 has to be included.

L226-229. Meat colour was different only for a* index; in this regard, the higher a* value detected in L+O meat could be linked in part to the higher pH2. Moreover, the comment regarding the highest L* and the lowest b* can be removed.

L234-235. This effect of linseed should be here better related to the enrichment of meat in unsaturated FA, as specified on L272-274.

L243. The comparison with Fusaro et al. is not appropriate, since that paper studied suckling lambs.

L245-246. These differences were very negligible, however the lower lipid content with L+O diet (p=0.055) was in line with the lower dissectible fat percentage (table 6); although this latter difference was not significant, this aspect could be mentioned.

L254. It is known that the stearic acid does not show a hypercolesterolemic effect, and also its level with L+O diet was not lower than with the control diet. The suggestion is to exclude the stearic acid for this comment.

L261-263. Giannico et al. compared the use of linseed oil and extruded linseed with soybean oil, as in this experiment, this aspect could be specified.

L306. In the mentioned papers (50-52), the different meat flavour is attributed to the pasture, so there is no analogy with this experiment.

L313. Based on the results on table 8, the sensory parameters were not homogeneous, as well as the fat content of meat tended to be not uniform (table 8). Other factors rather than fat could be responsible of the higher scores attributed to L+O meat.

Author Response

RESPONSES TO THE REVIEWER'S COMMENTS

Dear reviewer, corrections into the manuscript are highlighted in yellow.

Specific remarks and suggestions

L12. “sensory quality” should be substituted by “sensory attributes”.

we rewrote the sentence

L17. “was not affected” instead of “has not affected”.

we rewrote the sentence

L21-23. This sentence should be rearranged: “The combination of nutritional and sensory traits with properties related to human health….”.

we rewrote the sentence

L61. …because they are… (“of” has to be removed).

done

L67. This assertion means that the authors hypothesize that the effect of oregano supplementation could be different in the meat from Gentile di Puglia lambs than in meat from lambs of other breeds. A more convincing motivation could be that there is few information about the effects of combination of oregano with linseeds on lamb meat, taking into account that oregano could be able to improve oxidative stability of meat enriched in unsaturated FA due to the use of linseeds in the diet.

We accepted the suggestion and we rewrote the sentence

L72. …the dramatic reduction of its population…

we rewrote the sentence

L88. The study was carried out….

we rewrote the sentence

L91. The colon after “as” has to be removed.

done

L92. Were the lambs assigned to the treatments in a homogeneous way? This aspect has to be reported.

we rephrased the sentence

L126. Net energy is more suitable for ruminant species.

We found some reference in which is used the metabolizable energy:

Atti, N.; Methlouthi, N.; Saidi, C.; Mahouachi, M. Effects of extruded linseed on muscle physicochemical characteristics and fatty acid composition of lambs. J. Appli. Anim. Res. 2013, 41 (4), 404-409. DOI: 10.1080/09712119.2013.792730

Omar, C.A.; Yousif, A.N.; Arif, M.K.; Zahir, H.G. Effect of ground flaxseed on the carcass characteristics of Karadi male lambs. Iraqi J. Vet. Sci. 2019, 33, 1, 93-98. DOI: 10.33899/ijvs.2019.125517.1039

Marino, R.; Caroprese, M.; Annicchiarico, G.; Ciampi, F.; Ciliberti, M.G.; della Malva, A.; Santillo A.; Sevi, A; Albenzio M. Effect of dietary supplementation with Quinoa seed and/or Linseed on immune response, productivity and meat quality in Merino derived lambs. Animals 2018, 8, 204. DOI: 10.3390/ani8110204

Table 2. The sum of fatty acids of C diet is lower than 50%. What about the other fatty acids?

We wanted to list only the fatty acids that recorded the greatest differences, to make the table more readable

Were DPA and DHA present in the diets at these levels?

We apologize but there was a mistake, we corrected it

L171-176. The use of a 9-point hedonic scale is repeated more times.

We corrected the sentence

L190.  …diets did not influence… (the use of “no” with a verb, here and in many cases in the text, is not correct).

We corrected the sentence in line 190 and in the text

L191-193. The comparison with Leal et al. seems not to be particularly pertinent, it does not explain the lack of significant differences in the lambs ADG.

We preferred to delete the comment.

Whereas in the comparison with Lestingi et al., the authors should specify that lambs of the same Gentile di Puglia breed were used also in that research.

We corrected the sentence

Table 4. What about the feed intake of lambs? Although it can be calculated by the feed conversion ratio (kg/d has to be removed), the statistical comparisons could be showed.

We did not want to insert the data as the recorded feed intake have been very variable between the subjects of the same group

In addition, the “p value” is never higher than 1; the levels of significance >1 have to be checked also in the tables 5 and 7.

We apologize for mistake, we corrected it

Table 5. The letters used for means comparisons have to be reported for all three compared means (see abdominal region). Also the other tables have to be checked for this aspect.

We have inserted the necessary letters

L214. This sentence should be moved in the previous paragraph.

We changed the paragraph title

Table 6. A note for pH1 has to be included.

We added a note for pH1

L226-229. Meat colour was different only for a* index; in this regard, the higher a* value detected in L+O meat could be linked in part to the higher pH2. Moreover, the comment regarding the highest L* and the lowest b* can be removed.

We corrected the sentence

L234-235. This effect of linseed should be here better related to the enrichment of meat in unsaturated FA, as specified on L272-274.

we rephrased the sentence

L243. The comparison with Fusaro et al. is not appropriate, since that paper studied suckling lambs.

In our studies on the effect of the integration of linseed and linseed + oregano on the goats, we found effects on the kids’ meat, for this reason we considered appropriate the comparison with Fusaro et al.

L245-246. These differences were very negligible, however the lower lipid content with L+O diet (p=0.055) was in line with the lower dissectible fat percentage (table 6); although this latter difference was not significant, this aspect could be mentioned.

we rephrased the sentence

L254. It is known that the stearic acid does not show a hypercolesterolemic effect, and also its level with L+O diet was not lower than with the control diet. The suggestion is to exclude the stearic acid for this comment.

We accepted the suggestion and we delated the stearic acid in the sentence

L261-263. Giannico et al. compared the use of linseed oil and extruded linseed with soybean oil, as in this experiment, this aspect could be specified.

We added the specific note

L306. In the mentioned papers (50-52), the different meat flavour is attributed to the pasture, so there is no analogy with this experiment.

The judges who evaluated the meat of our work showed a difference according to the feed integration of the lambs; the same thing happened in the cited papers. Although the feeding was different we used these referee to highlight how judges might notice these differences.

L313. Based on the results on table 8, the sensory parameters were not homogeneous, as well as the fat content of meat tended to be not uniform (table 8). Other factors rather than fat could be responsible of the higher scores attributed to L+O meat.

We preferred to cancel comment

Reviewer 2 Report

The authors present a study on the effects of two diet supplements on several meat quality parameters in lambs. The data gathered in this study originates from a sufficiently large sample size for testing some of the predictions (which, unfortunately, are not spelled out anywhere in the manuscript); however, there are major flaws in the data analyses and presentation. I therefore cannot recommend acceptance for publication at this point. Please see my specific comments below.

Major comments
Language use is by no means up to par and must be substantially improved before publication. Mostly, this refers to simple grammatical errors and poor wording, however, at times, it is obvious that the meaning is entirely misconstrued or ambiguous. As it stands, the language quality of this manuscript is not acceptable for publication. I strongly recommend language editing by a native speaker. Please note that in my minor comments, I only corrected very few of the many errors.

There are considerable flaws in the presentation of this manuscript, including unjustified statements and, what is more concerning, inaccurate/inadequate referencing to support such claims. In the first sentence, the authors basically claim that red meat is an important if not almost the only necessary part of the human diet - this is completely overstated and utterly untrue. Whats more, the reference cited here is a study on diet supplements of lambs, which has nothing to do with the content of this sentence. I suppose reference [2] was meant to be cited here as this is a review on red meat in the diet; however, the main conclusion of this review is that red meat consumption should be limited or reduced. As such, the interpretation by the authors of the current study seems concerningly inadequate.

The claim in line 56f (“increased PUFA level may limit shelf-life of meat, because they are more prone to oxidation”) is not supported or even touched by reference [13] which examines the correlation of polyunsaturated fatty acids and the risk of prostate cancer.
Line 60ff - the authors claim that consumer consider certain herbs to be safe and healthy, yet the reference they provide is a study on effects of entirely different compounds on chicken meat (nothing to do with consumer perception of herbs).

I am missing clear predictions (i.e., what effect of which dietary compound on what meat parameter was expected?).

The introduction makes a compelling and valid point regarding the relevance of this study and of marketable products of this particular breed. To improve the narrative of this manuscript, I would suggest moving the respective section to the beginning of the introduction, and from there move on to red meat as a diet component and how its quality may be affected using feed supplements.

Statistics -as it stands, this crucial part of the study is not reproducible which makes it hard to judge; however, the testing procedure seems flawed. It is unclear what predictors (or other specifications) were applied in the ‘final’ model - which brings up the question which other models were fitted and whether the information criteria of those models were weighed against each other? If so, why are these results not presented? The authors claim that “initial body weight” was removed from the model as it had no significant effect - I argue that whatever the effect, logically, this is an important control variable (that thus must be controlled for in a GLM!), and the argument that it is non-significant and was therefore removed is not valid.
In any case, simply quoting some p-values as results of GLMs is not acceptable; the model parameters and statistics must be quoted, otherwise it is not possible to judge validity; I suggest the authors consult with a statistician or read up on how to properly report such results (e.g., https://doi.org/10.1111/2041-210X.12577). Also, please note that the term “GLM” should stand for “generalized linear model” (not ‘general linear procedure’).

The Results & Discussion section must be revised thoroughly. The point of a discussion section is to guide the reader through new insights with three crucial aspects in mind, i.e., interpretation of the findings, potential implications, and recommendation for further research. Please note that it is insufficient to simply present findings and then mention what previous studies may or may not have found. This is also where having formulated proper predictions is helpful as these can be discussed here with the reader aware of the point that is being made.

Inadequate use of abbreviations -
line 18 The abbreviations SFA and n-3 PUFA were not introduced (nor are they used again in the summary, thus the spelled-out terms should be used)
line 47 the abbreviation UFA does not occur again and should thus be removed here.
line 52 the abbreviation SFA was already introduced in line 46; remove the spelled-out term here
line 53 the abbreviation FA was not introduced
Table 4 “BW” not defined
line 110 the term “pH24” is introduced here, but does not occur again; however, in line 225, the term “pH2” occurs without having been defined anywhere. In the legend of Table 7, a mix is used (“pH2 at 24 h post-mortem”) - should be “pH2 (pH 24 h post-mortem)”.
line 165 TBARS or T-BARS? Use one term consistently.
line 182 The abbreviation “GLM” is not used again and should thus not be introduced.

Punctuation - the authors should use commas after introductory clauses, which would considerably improve readability.

line 50 the correct term is “polyunsaturated fatty acids” (remove the space between poly and unsaturated).

line 191 the intended meaning of this phrase is entirely unclear. please clarify.
line 194 no effect on “efficiency” - what does this refer to?
line 170f what is meant by “indicated intensity on a 9-cm linear scale”? I don’t understand how cm is a measurement of sensation.

Table 3 please explain more clearly the difference between ’succulence’ and ‘juiciness’, as these seem to be identical concepts in this context.
Tables 5, 6, 7 (legends) it should be “a vs. b” (not a-b)
Table 6 The content is obscure and confusing - what are the units in lines “Lean” and “Bone”?
Tables 5 and 9 I suggest showing significant p-values in bold print, for better visibility.

Minor comments
Summary
line 13 add “an” (…quality of an Italian…)
line 17 “was not affected” rather than “has not affected”
line 17 “Supplementation of the diet with” (not “supplementation of … to diet”)

Abstract
line 27 light lambs are consumed - heavy ones are not?
line 29 Numerals should be spelled-out at the beginning of a sentence
line 30 what do you mean by ‘homogenous group’? Equally sized groups?
line 33 The meaning of “L1” is unclear at this point.
line 35 The meaning of n-3 and n-6 is unclear here.

Intro
line 44 acids (plural)
line 46 long, not large

Methods
line 88 add “was” (the study was carried out…)
several instances - add a space between numbers and the unit “°C”
line 105 Average Daily Gain - use lower-case letters
line 108 the percent symbol is part of the numeral (not a unit), thus its “80%-82%” (not “80-82%”)
line 112 add “USA” after “RI”
line 122 add a hyphen (“a 1-mm screen”)
line 122 ff the listed items are not “procedures” but components/parameters/compounds/etc.
line 137 for consistency, either spell out the American states or use abbreviations; currently, its a mix.
line 186 use spaces in p-value notations consistently (“P < 0.05”); same in table legends and in numerous other instances.
line 186 “unless otherwise stated” - is a different threshold of significance used anywhere in your manuscript? Not as far as I can tell; please remove this redundant phrase.
line 199 remove the period after the subheader
line 264 remove “(P > 0.05)” - this is entirely redundant.

Author Response

RESPONSES TO THE REVIEWER'S COMMENTS

Dear reviewer, corrections into the manuscript are highlighted in green. As you have requested, we changed the introduction paragraph and we have tried to correct all possible errors. Furthermore, we apologize for any possible oversights and errors

  • In the first sentence, the authors basically claim that red meat is an important if not almost the only necessary part of the human diet - this is completely overstated and utterly untrue. Whats more, the reference cited here is a study on diet supplements of lambs, which has nothing to do with the content of this sentence.

We have changed the introduction

  • I suppose reference [2] was meant to be cited here as this is a review on red meat in the diet; however, the main conclusion of this review is that red meat consumption should be limited or reduced. As such, the interpretation by the authors of the current study seems concerningly inadequate.

We have changed the introduction

  • The claim in line 56f (“increased PUFA level may limit shelf-life of meat, because they are more prone to oxidation”) is not supported or even touched by reference [13] which examines the correlation of polyunsaturated fatty acids and the risk of prostate cancer.

We change the reference with:

Wood, J.D., Richardson, R.I., Nute, G.R., Fisher, A.V., Campo, M.M., Kasapidou, E., Sheard, P.R., Enser, M. Effects of fatty acids on meat quality: a review. Meat Sci. 2004, 66, 21-32. DOI: 10.1016/S0309-1740(03)00022-6

  • Line 60ff - the authors claim that consumer consider certain herbs to be safe and healthy, yet the reference they provide is a study on effects of entirely different compounds on chicken meat (nothing to do with consumer perception of herbs).

We change the reference with:

Ahmad, S.R., Gokulakrishnan, P., Giriprasad R., Yatoo A. Fruit-based natural antioxidants in meat and meat products: a review. Critical Reviewes in Food Science and Nutrition, 2015, 55: 1503-1513. DOI: 10.1080/1040083998.2012.701674

Vital, A.C.P., Guerrero, A., Carvalho Kempinski, E.M.B., de Oliveira Monteschio, J., Sary, C., Rogelio Ramos, T., del Mar Campo, M., do Prado, I.N. Consumer profile and acceptability of cooked beef steaks with edible and active coating containing oregano and rosemary essential oils. Meat Sci. 2018, 143, 153-158, DOI: 10.1016/j.meatsci.2018.04.035.

I am missing clear predictions (i.e., what effect of which dietary compound on what meat parameter was expected?).

We hope that in the new form of introduction, our predictions will be clearer

The introduction makes a compelling and valid point regarding the relevance of this study and of marketable products of this particular breed. To improve the narrative of this manuscript, I would suggest moving the respective section to the beginning of the introduction, and from there move on to red meat as a diet component and how its quality may be affected using feed supplements.

We reformulated the introduction

Statistics -as it stands, this crucial part of the study is not reproducible which makes it hard to judge; however, the testing procedure seems flawed. It is unclear what predictors (or other specifications) were applied in the ‘final’ model - which brings up the question which other models were fitted and whether the information criteria of those models were weighed against each other? If so, why are these results not presented? The authors claim that “initial body weight” was removed from the model as it had no significant effect - I argue that whatever the effect, logically, this is an important control variable (that thus must be controlled for in a GLM!), and the argument that it is non-significant and was therefore removed is not valid.
In any case, simply quoting some p-values as results of GLMs is not acceptable; the model parameters and statistics must be quoted, otherwise it is not possible to judge validity; I suggest the authors consult with a statistician or read up on how to properly report such results (e.g., https://doi.org/10.1111/2041-210X.12577). Also, please note that the term “GLM” should stand for “generalized linear model” (not ‘general linear procedure’).

We apologize but unfortunately there was an error as the statistical analysis paragraph has not been updated. We changed it with the correct version.

The Results & Discussion section must be revised thoroughly. The point of a discussion section is to guide the reader through new insights with three crucial aspects in mind, i.e., interpretation of the findings, potential implications, and recommendation for further research. Please note that it is insufficient to simply present findings and then mention what previous studies may or may not have found. This is also where having formulated proper predictions is helpful as these can be discussed here with the reader aware of the point that is being made.

Inadequate use of abbreviations -
line 18 The abbreviations SFA and n-3 PUFA were not introduced (nor are they used again in the summary, thus the spelled-out terms should be used)

We added the spelled-out terms
line 47 the abbreviation UFA does not occur again and should thus be removed here.

Being the first time you meet this term we prefer to leave the abbreviation in brackets
line 52 the abbreviation SFA was already introduced in line 46; remove the spelled-out term here

done
line 53 the abbreviation FA was not introduced

we introduced the spelled-out
Table 4 “BW” not defined

we added a note
line 110 the term “pH24” is introduced here, but does not occur again; however, in line 225, the term “pH2” occurs without having been defined anywhere. In the legend of Table 7, a mix is used (“pH2 at 24 h post-mortem”) - should be “pH2 (pH 24 h post-mortem)”.

we apologize but there was a writing error
line 165 TBARS or T-BARS? Use one term consistently.

we have corrected all terms
line 182 The abbreviation “GLM” is not used again and should thus not be introduced.

We corrected it

Punctuation - the authors should use commas after introductory clauses, which would considerably improve readability.

We corrected it

line 50 the correct term is “polyunsaturated fatty acids” (remove the space between poly and unsaturated).

We corrected it

line 191 the intended meaning of this phrase is entirely unclear. please clarify.

We reworded the sentence
line 194 no effect on “efficiency” - what does this refer to?

we preferred to delete the entry
line 170f what is meant by “indicated intensity on a 9-cm linear scale”? I don’t understand how cm is a measurement of sensation.

we aligned with the 9-point scale

Table 3 please explain more clearly the difference between ’succulence’ and ‘juiciness’, as these seem to be identical concepts in this context.

Succulence is the first perception of the amount of liquid liberated by the sample while juiciness is the same perception after 5th bite.

Tables 5, 6, 7 (legends) it should be “a vs. b” (not a-b)

We corrected it

Table 6 The content is obscure and confusing - what are the units in lines “Lean” and “Bone”?

We added the unit in lines

Tables 5 and 9 I suggest showing significant p-values in bold print, for better visibility.

Bold in p-values is not allowed by author instructions

Minor comments – we corrected all in the text
Summary
line 13 add “an” (…quality of an Italian…)
line 17 “was not affected” rather than “has not affected”
line 17 “Supplementation of the diet with” (not “supplementation of … to diet”)

Abstract
line 27 light lambs are consumed - heavy ones are not?

In the South of Italy only light lambs are consumed, heavy ones are consumed in North Italy and North Europe
line 29 Numerals should be spelled-out at the beginning of a sentence
line 30 what do you mean by ‘homogenous group’? Equally sized groups? It is explained at lines 91-94
line 33 The meaning of “L1” is unclear at this point.
line 35 The meaning of n-3 and n-6 is unclear here.

Intro
line 44 acids (plural)
line 46 long, not large

Methods
line 88 add “was” (the study was carried out…)
several instances - add a space between numbers and the unit “°C”
line 105 Average Daily Gain - use lower-case letters
line 108 the percent symbol is part of the numeral (not a unit), thus its “80%-82%” (not “80-82%”)
line 112 add “USA” after “RI”
line 122 add a hyphen (“a 1-mm screen”)
line 122 ff the listed items are not “procedures” but components/parameters/compounds/etc.

method 934.01 and other are AOAC procedures to determinate the components amount
line 137 for consistency, either spell out the American states or use abbreviations; currently, its a mix.
line 186 use spaces in p-value notations consistently (“P < 0.05”); same in table legends and in numerous other instances.
line 186 “unless otherwise stated” - is a different threshold of significance used anywhere in your manuscript? Not as far as I can tell; please remove this redundant phrase.
line 199 remove the period after the subheader
line 264 remove “(P > 0.05)” - this is entirely redundant.

Round 2

Reviewer 2 Report

I commend the authors’ effort to improve their manuscript. The structure of the introduction was substantially improved and reads much better now. Also the experimental methods are much clearer. The manuscript has certainly been improved substantially, and I still think this is a sound study with interesting results generated from valid data; however, the statistics (methods and results) are still not sufficiently clear.

In the methods section, I need the authors to specify the structure of their GLM(s) - body weight was the response variable, and dietary treatment was a predictor I presume? What else? Please clarify this in detail so that anyone who has access to your raw data would be able to simply reproduce your results. What is meant by ‘experimental error’ (line 184f)? Overall, the statistics results, and particularly the table content, are still unclear to me - where do these p-values come from? Do they refer to the effect of the dietary treatment in the model? It does not appear so, as the same p-values shown in the tables also occur in the text as what seems to be the results of pairwise post hoc tests? Again, if so, what about the actual model results? As I noted in the previous round, the reporting of GLM results must be substantially revised, unfortunately the authors have chosen not to respond to this comment or to my question regarding inclusion/exclusion of the predictor “initial body weight” in the model. 
The indication of statistical significance in the tables has become confusing - now double letters (ab) are used which seem to indicate something different than stated in the text. E.g., table 6: ‘dissectible fat” suggests a difference between L (a) and L+O (b), which does not seem to be right, according to the text; according to the paragraph above, this should be C (a), L (b), L+O (b), as C differs significantly from L and L+O, but no significant difference occurs between L and L+O. Is this right? Double letters are not helpful but confusing, it is unclear what (a) (ab) (b) is supposed to indicate in this context. If two treatments differ from one, use e.g. (a) (a) (b); if all three treatments differ from each other, use (a) (b) (c); this should be clear. Please revise all respective tables accordingly and clarify in the legends that “Different (lower-case) letters indicate statistical significance”, etc.  
In the results, please clarify what is meant by all the ‘percentages’ - I presume this refers to the proportion of body weight? 

Minor comments - English language use should still be improved, below please find a few (exemplary!) comments which are by now means exhaustive.
line 16    remove “either in” (“…effect on either in…”
line 17     remove “the” (“the meat chemical…”)
line 19    increases
line 20    “and so were the lamb …”
line 21    “are compatible”
line 27    on instead of into
line 34    “no” rather than “not a”
line 54    perhaps “favorable proportion” rather than ‘good range’
line 81    add “conducted” (this study was …)
line 92    add a comma after “each”; better “regarding” than “ for”
line 105    better “each week” than weekly
line 106    “until” rather than since
line 111    “in” (not on the muscle)
line 199    lean what? meat? this isolated adjective occurs several times…
line 156    space between 135 and °C
line 174f    perhaps better “between tasting two samples”
line 183    remove ANOVA; the term does not occur again and is not informative here
line 190     “were found” rather than are showed
line 194    “even though” rather than even if
line 201    “showed” rather than recorded (same in line 207 and in several other instances)
line 203    “studies”, not researchers
line 207    please use the exact p-value rather than just the threshold
Table 5    use fleece (as above) rather than wool
line 222    remove the period after the header
line 228f    remove “in the results” and “it” in line 229
line 229     use “and” instead of the comma
line 231    “in” not for
line 273    “re showed in [8] and [45] papers” …please revise.

Author Response

Dear referee thank you for the right and precise suggested corrections. We hope that this new version of the text will meet your expectations

We hope that with the revision of statistical analysis paragraph, all the procedures will be clear.

What is meant by ‘experimental error’ (line 184f)? was the residual error

Overall, the statistics results, and particularly the table content, are still unclear to me - where do these p-values come from? Do they refer to the effect of the dietary treatment in the model? Yes, they do

 It does not appear so, as the same p-values shown in the tables also occur in the text as what seems to be the results of pairwise post hoc tests? We have rewrote as the statistical paragraph as the results

The indication of statistical significance in the tables has become confusing - now double letters (ab) are used which seem to indicate something different than stated in the text.

No, it doesn’t. Nowadays it is used to put double letters (ab) close to data that have no statistical significance to indicate that it has not difference with “a” or with “b”

E.g., table 6: ‘dissectible fat” suggests a difference between L (a) and L+O (b), which does not seem to be right, according to the text; we corrected the sentence

 according to the paragraph above, this should be C (a), L (b), L+O (b), as C differs significantly from L and L+O, but no significant difference occurs between L and L+O. Is this right? Yes, it is right

Double letters are not helpful but confusing, it is unclear what (a) (ab) (b) is supposed to indicate in this context. If two treatments differ from one, use e.g. (a) (a) (b); if all three treatments differ from each other, use (a) (b) (c); this should be clear. Please revise all respective tables accordingly and clarify in the legends that “Different (lower-case) letters indicate statistical significance”, etc.  

In the first version of the article the double letters (ab) were not there; they were added as required by the reviewer n. 1. In recent years their use is spreading into similar papers, several authors use this form.

Unfortunately, if reviewer 1 say to put (ab) in the table and reviewer 2 say to delete them, we don’t know what to do…

In the results, please clarify what is meant by all the ‘percentages’ - I presume this refers to the proportion of body weight? 

In line 206 is specified that percentage of abdominal region is referred to the half carcass weight, it is also written in the note n. 4 of table 5; while the percentage of dissection of leg and loin are referred to the weight of each cut. Maybe this was unclear, so we added “cut weight” on table title

line 199    lean what? meat? this isolated adjective occurs several times…

Usually in literature, when someone dissects a carcass cut he speaks of lean, fat and bone. Lean is composed of all the muscles present in the cut, so it become a subject, it is not an adjective
line 222    remove the period after the header

I apologize but I did not understand the request

Round 3

Reviewer 2 Report

The statistics have been clarified sufficiently, and other errors were corrected.

Author Response

thank you!